# Exploring traumatic childbirth: Associations between obstetric violence indicators and perinatal posttraumatic stress

Maria Vega-Sanz, Amaia Halty*, Sofía Goñi-Dengra, Carlos Pitillas, Ana Berástegui

University Institute of Family Studies, Pontifical Comillas University, Madrid, Spain

* ahalty@comillas.edu

## Abstract

### Objective

The childbirth experience can be traumatic for women, with negative repercussions on their mental health, mother-child bonding, and subsequent infant development. The aim of this study is to analyze the negative birth experience, through indicators of obstetric violence (OV), as a risk factor for the development of Perinatal Posttraumatic Stress (P-PTS) in early postpartum. Additionally, we seek to explore the buffering impact of other variables on the development of P-PTS symptomatology.

### Methods

A total of 236 postpartum women were surveyed online, between the fourth and sixth week postpartum, assessing thirteen indicators of OV using the Questionnaire on Birth Conditions. We also utilized the Post-traumatic Stress Symptom Checklist and Multidimensional Scale of Perceived Social Support, alongside collecting sociodemographic, pregnancy and delivery conditions. Mean differences, correlations, and regression analyses were performed.

### Results

Women exposed to OV have a higher risk of developing P-PTS symptoms. Increased risk was noted in those exposed to staff's ironic comments, undergoing medical procedures without prior information, or those who were made to feel guilty for childbirth outcomes. Moreover, early postpartum skin-to-skin contact and perceived social support from friends and family served as protective factors against P-PTSS development.

### Conclusions

Postpartum traumatization may not solely stem from threats to physical integrity or survival but also from experiences of inferiority, inadequacy, loss of dignity, or

**Data availability statement:** The dataset analyzed during the current study is available in Zenodo. https://doi.org/10.5281/zenodo.15024158.

**Funding:** Universidad Pontificia Comillas

**Competing interests:** The authors have declared that no competing interests exist.

dehumanizing treatment. These findings underline the necessity for healthcare professionals to enhance the quality of care during childbirth, to maximize immediate skin-to-skin contact, and been aware and actively seeking social support for women.

## Introduction

Childbirth can be experienced by women as a traumatic event that may result in perinatal posttraumatic stress symptoms (P-PTSS). It is important to note that the perinatal period encompasses not only childbirth but also pregnancy and the immediate postpartum phase. Following the recognition that some experiences during pregnancy and childbirth are sufficiently traumatic to lead to clinical conditions of Post Traumatic Stress Disorder (i.e., PTSD) [1], the term "Perinatal Post Traumatic Stress Disorder" (P-PTSD) was introduced to describe these clinical conditions. P-PTSS refers to the development of posttraumatic symptoms derived from or related to events occurring throughout the perinatal period, and it is distinct from postpartum PTSD or symptoms that arise solely after childbirth. These symptoms often include feelings of fear, terror, hopelessness, or helplessness [1–5].Some studies have found that around 19.7–45.5% of women perceive their birth as traumatic [6] and around 30% of women meet several diagnostic criteria for P-PTSD after childbirth. [7]. Furthermore, it has been found that the current prevalence of perinatal post-traumatic stress disorder (P-PTSD) in Spain is 11.1% [4]. The incidence of P-PTSD Increases up to 25% in cases of stillbirth and up to 30–35% in cases of infant death during NICU admission it increases to 30–35% [8]. The clinical presentation of P-PTSD, as for PTSD, is primarily characterized by nightmares, flashbacks, irritability, guilt, and attempts to avoid thinking or talking about the difficult event [8].

A great deal of research has been conducted on the risk factors that contribute to triggering P-PTSS. The presence of psychological [9] and medical complications during pregnancy such as pregnancy induced hypertension [10], or preterm births [11] would act as a risk factor for the subsequent development of P-PTSS [10]. In relation to the birth experience, the most frequent factors are the presence of medical complications such as the performance of an emergency caesarean section [12], the quality of medical care received [7,13], medical complications in the newborn [13], and a low birth satisfaction [14]. In this context, there is increasing interest in exploring the potential role of Obstetric Violence (OV) as an additional risk factor to the development of P-PTSS. Some indicators of obstetric violence that would also be associated with the development of P-PTSS after childbirth but have been underexplored in the scientific literature [3]. Therefore, the present study specifically aims to examine how indicators of OV during childbirth may contribute to the development of P-PTSS.

### Obstetric violence

Obstetric Violence (OV) encompasses a range of practices during childbirth that violate women's autonomy, dignity, and rights. It can manifest physically through

medical interventions performed without consent or with insufficient information, and psychologically, through degrading or infantilizing treatment by healthcare providers [15]. These experiences can lead to feelings of vulnerability, guilt, and loss of control, impacting maternal well-being and birth outcomes [3,5,16].

Physically, OV includes procedures carried out without proper consent or under explicit refusal, such as episiotomies, repeated vaginal examinations by different professionals, the Kristeller maneuver, medically unnecessary acceleration or inhibition of labor, and the forced adoption of specific birthing positions without clinical justification [16]. The WHO does not recommend these practices due to their potential negative consequences, including chronic pain, dyspareunia, perineal trauma, and urinary or fecal incontinence [17].

Psychologically, OV occurs when women are subjected to dehumanizing, dismissive, or authoritarian behavior by healthcare personnel. This includes being ignored, treated with sarcasm, denied the right to express emotions, or made to feel guilty for childbirth [3,5,16]. In this context, the psycho-affective needs of women—such as emotional validation, empathetic communication, and a supportive environment—are severely compromised. This neglect intensifies feelings of vulnerability, loss of agency, and isolation, thereby exacerbating the adverse psychological effects and increasing the risk of developing P-PTSS. Additionally, a lack of respect for the woman's birth plan, as well as coercion in decision-making, further contributes to these lasting emotional consequences [16]Beyond its physical and psychological effects, OV also involves moral and epistemic injustice. The study by van der Waal et al. [15] highlighted the moral priority often given to the baby over the mother's well-being. Additionally, when mothers provide consent, it is sometimes obtained illegitimately, potentially involving forms of coercion or misinformation, leading to epistemic injustice.

## Obstetric violence and perinatal post-traumatic stress

Previous studies have found that the indicators of OV most associated with the subsequent development of P-PTSS are non-compliance with the birth plan by the health care team and emergency caesarean deliveries [3,5]. However, other studies also indicate that vaginal deliveries carried out when the woman preferred a caesarean section are also associated with the development of P-PTSS [18].

In addition to the fear that may arise from perceiving threats to one's own life or that of the infant, feelings of shame commonly manifest during childbirth [19]. Beyond the perinatal setting, a substantial body of empirical evidence links PTSD to experiences of shame across various traumatic contexts, including interpersonal violence, physical or sexual child abuse, and war [20].

When women undergo the childbirth experience and encounter feelings of shame alongside emotions such as loneliness, disrespect, insignificance, or ignorance, a psychobiological reaction is triggered, detrimentally impacting the childbirth process itself [21,22]. Specifically, the occurrence of these emotions has been associated with heightened pain perception, reduced pain tolerance, increased demand for epidural anaesthesia, prolonged labour duration [23], and a higher risk of caesarean section [24].

It has been demonstrated that P-PTSS conditions can have far-reaching implications beyond solely affecting the well-being of mothers. These repercussions extend to relational issues with the baby, such as disrupting breastfeeding [25], maternal-infant bonding [25–27], difficulties in performing caregiving tasks for the baby [27], and negatively affecting their perceived self-efficacy [28]. All these factors have an impact on the child's own development, including neurological and cognitive development [29,30], emotional regulation [31], and social adjustment [32], so addressing them can avoid negative consequences for the whole family.

Despite all the above, a protective factor against the development of P-PTSS after a traumatic birth is the opportunity to have skin-to-skin contact with the newborn [16,33]. Studies have identified that skin-to-skin contact after a traumatic birth experience reduces post-traumatic stress symptomatology through a reduction in feelings of guilt and fear related to childbirth [33]. In addition, skin-to-skin contact has been associated with the establishment of more successful breastfeeding [34], which promotes maternal confidence and reassurance. This would help to reduce the mother's emotional

distress associated with an adverse birth experience [35] and improve the quality of the mother-baby bond [36]. Moreover, it is important to consider that perceived social support may act as a protective factor against these adversities, especially when it comes from the family, the partner, and the healthcare team [12]. Social support affects cognitive evaluations and has been associated with less pessimistic views about the future [37]. Consequently, women lacking sufficient support from friends and family may be more susceptible to feeling anxious about childbirth and viewing their childbirth experiences negatively [10].

Given the relevance of childbirth as a bridging event between pregnancy and the postpartum period, it is necessary to advance in the study of the psychological implications of it being experienced as a traumatic event. The present study aims to identify specific indicators of OV during childbirth that may contribute to the development of P-PTSS. A deeper understanding of OV during childbirth could be a great step forward in the prevention of the appearance of P-PTSS and all the consequences derived from this. To exploring which indicators of OV may represent a risk factor for the development of P-PTSS after childbirth, we will also examine other variables related to pregnancy, childbirth, and postpartum that have been identified as significant in previous empirical research.Specially, a linear regression model will be performed for P-PTSS, encompassing factors from pregnancy (medical problems during pregnancy), childbirth (type of delivery, medical problems during delivery), OV items from the childbirth conditions questionnaire, and perceived social support during the immediate postpartum period (from friends, family, and significant other).

## Method

### Participants and procedure

A non-discriminatory exponential chain demonstration (i.e., snowball sampling technique) was performed to recruit the sample (N = 236). The dissemination of the questionnaires was carried out through social networks, and participants were recruited via these platforms. They completed the questionnaires online through a Microsoft Forms survey. Sample recruitment began on 7 February 2023 and ended on 5 May 2023. The inclusion criteria for the study were being a woman over 25 years of age, residing in Spain, and being between the fourth and sixth week postpartum.

In Spain, there are different healthcare services for pregnancy and childbirth: the public health service, private medical insurance, and entirely private healthcare. The latter two options offer greater flexibility in choosing the hospital and doctor. Most births take place in public or private hospitals, with the use of birthing centers being very uncommon and home births extremely rare. Delivery can be either vaginal or by cesarean section, depending on the circumstances and healthcare professionals' preferences. During labor, women are attended by midwives and obstetricians, although the role of the doula is not yet established. Typically, the presence of a companion is allowed, except for cesarean deliveries, which vary by hospital. Options for pain control are offered, with pharmacological methods being the most prevalent nowadays, especially the use of epidurals.

The mean age of the final sample was 32.51 (SD = 3.76), 94% of the participants completed secondary education, and 77% gave birth in a public hospital. 91% of the sample were first-time mothers and 99% of mothers gave birth to only one baby.

General aspects of the delivery are shown in Table 1. Informed consent was obtained from each participant in written form, using an online form. The study protocol was approved by an ethics committee (reference hidden) (ref. no. 45/22–23).

### Measures

Data was collected through online evaluation surveys using Microsoft Forms. The surveys included questions on socio-demographic and general aspects of pregnancy (such as medical complications during pregnancy) and childbirth (type of delivery, medical complications during childbirth, and whether skin-to-skin contact occurred after birth).The following questionnaires were used to assess risk factors for P-PTSS.

**Table 1. General aspects of childbirth.**

| Variable | Total Sample (N = 236) | |
|---|---|---|
| | N | % |
| Complications during pregnancy | | |
| Yes | 62 | 26.3 |
| Type of delivery | | |
| Natural | 183 | 77.5 |
| Scheduled cesarean section | 7 | 3 |
| Emergency caesarean section | 46 | 19.5 |
| Complications during delivery | | |
| Yes | 53 | 22.5 |
| Baby's gestational week at birth | | |
| Extremely preterm | 1 | 0.4 |
| Very preterm | 1 | 0.4 |
| Moderate preterm | 13 | 5.5 |
| Full term | 221 | 93.6 |

*Note. Extremely preterm = less than 28 weeks; Very preterm = 28–32 weeks; Moderate preterm = 32–37 weeks; Full term = equal to or more than 37 weeks.*

**Indicators of OV.** Questionnaire on Birth Conditions [38]. This questionnaire allows for the identification of some indicators of OV during childbirth [3,5,16]. It comprises 13 items (see Table 2 for details) and a dichotomous response scale (yes/no). The original authors proposed that if the woman answers affirmatively to only one of the items, she is considered to have suffered OV but, in this study, the items have been used as 13 independent indicators. Additionally, and to further clarify the information regarding the "procedures without information" item, exposure to the following procedures performed without providing information was assessed: vaginal exams; refusal to provide food; artificial rupture of membranes (AROM); episiotomy; instruction to always lie down during childbirth; intravenous catheter insertion; genital shaving; enema; abdominal compression (Kristeller maneuver). The questionnaire was used in its original Spanish version for the research, and the research team performed the English translation of these items to facilitate better understanding for this paper's publication. The items were translated by bilingual colleagues from the research team.

**P-PTSS.** The Post-traumatic Stress Symptom Checklist (PCL-5, [39]) comprises 20 items and a Likert-type response scale (5 response options). This scale has a validated Spanish adaptation and also evidence of its cross-cultural validation [40]. Although it was originally developed for assessing trauma-related symptoms in the general population, it has been specifically validated for evaluating traumatic childbirth experiences [41]. It was specified that mothers should complete this questionnaire based on their childbirth experience. The internal consistency in this study at 4–6 weeks at postpartum was $\alpha = .95$.

Since no defined cutoff point exists, the authors suggest classifying the questionnaire items according to the symptom clusters established in the DSM-5. The items were grouped into the following clusters: B (items 1−5), C (items 6−7), D (items 8−14), and E (items 15−20). Each participant's responses were then evaluated to determine how many items they endorsed within each cluster, following the DSM-5 diagnostic rule, which requires at least 1 item from cluster B, 1 from cluster C, 2 from cluster D, and 2 from cluster E to meet PTSD criteria.

Participants who met this criterion were categorized as having a clinical level of P-PTSS, even though the temporal criterion of a minimum symptom duration of six months for a formal P-PTSS diagnosis had not yet been met. Those who endorsed symptoms in three out of the four symptom clusters were classified as having a subclinical level of P-PTSS.

This methodology was used to obtain the percentage of subjects presenting symptoms compatible with P-PTSS For the rest of the analyses, the sum of the items on the scale was used as the P-PTSS score.

**Table 2. Frequencies of items indicating of obstetric violence and t-test comparisons in P-PTSS.**

| | Yes (%)/ No (%) | Mean (sd) | t | gl | sig |
|---|---|---|---|---|---|
| 1. Did you feel that your labor was sped up or slowed down for the convenience of the healthcare staff? [¿sentiste que te aceleraron o ralentizaron el parto por conveniencia del personal sanitario?]} | 33.9 | 22.49 (18.89) | −4.59 | 125.53 | p<0.001 |
| | 66.1 | 11.60 (13.92) | | | |
| 2. Did trainees intervene during labor or childbirth without your consent? [¿intervinieron estudiantes en prácticas durante el trabajo de parto o el parto sin tu consentimiento?] | 17.8 | 24.02 (18.70) | −3.48 | 55.196 | p<0.001 |
| | 82.2 | 13.37 (15.45) | | | |
| 3. Did you feel that you were being infantilized (through the use of diminutives, nicknames, by making decisions without informing you, oversimplifying explanations, etc.) as if you couldn't understand what was happening or express your opinion? [¿sentiste que te infantilizaron (con diminutivos, sobrenombres, al tomar decisiones sin informarte, simplificando demasiado las explicaciones, etc.) como si no pudieras entender lo que pasaba u opinar?] | 22.5 | 27.78 (18.72) | −5.87 | 70.97 | p<0.001 |
| | 77.5 | 11.65 (13.96) | | | |
| 4. Did you find asking questions or expressing your fears or concerns difficult or impossible because you were not answered or answered badly? [¿te fue difícil o imposible preguntar o manifestar tus miedos o inquietudes porque no te respondían o lo hacían de mala manera?] | 16.1 | 31.80 (19.11) | −6.34 | 48.86 | p<0.001 |
| | 83.9 | 11.87 (13.74) | | | |
| 5. Did you feel neglected by the healthcare staff during labor? [¿Te sentiste desatendida por parte del personal sanitario durante el trabajo de parto?] | 20.8 | 28.08 (20.10) | −5.52 | 64.23 | p<0.001 |
| | 79.2 | 11.73 (13.49) | | | |
| 6. Were you forced to stay in bed, prevented from walking or changing position? [¿te obligaron a quedarte en cama, impidiéndote caminar o cambiar de posición?] | 29.2 | 25.11 (19.66) | −5.43 | 95.51 | p<0.001 |
| | 70.8 | 11.22 (16.16) | | | |
| 7. Do you think the healthcare staff made ironic, disparaging, or mocking comments about your behavior or feelings? [¿crees que el personal sanitario hacía comentarios irónicos, descalificadores o en tono de burla de tu comportamiento o tus sensaciones?] | 10.6 | 33.11 (18.64) | −6.42 | 234 | p<0.001 |
| | 89.4 | 13.01 (14.86) | | | |
| 8. Were you prevented from expressing your emotions (crying, screaming, laughing, etc.) during labor or childbirth? [¿crees que te impidieron expresar tus emociones (llorar, gritar, reírte, etc.) durante el trabajo de parto o parto?] | 8.1 | 35.90 (20.10) | −5.0 | 22.12 | p<0.001 |
| | 91.9 | 13.30 (14.78) | | | |
| 9. Please let us know if any of the following procedures were carried out without providing you with sufficient information *(See Instruments section for more details)* [Por favor, dinos si alguno de los siguientes procedimientos se realizó sin proporcionarte suficiente información] | 63.6 | 13.33 (7.50) | −2.22 | 11.07 | .048 |
| | 36.4 | 7.30 (9.90) | | | |
| 10. Did you feel threatened or insulted during labor or childbirth?. [¿te sentiste amenazada o insultada durante el trabajo de parto o el parto?] | 5.9 | 36.13 (21.30) | −3.99 | 14.97 | p<0.001 |
| | 94.1 | 13.88 (15.26) | | | |
| 11. At the time of delivery, were you prevented or hindered in your choice of birthing position? [en el momento del parto¿se te impidió o dificultó elegir posición para parir?] | 30.1 | 24.80 (19.08) | −5.62 | 105.92 | p<0.001 |
| | 69.9 | 11.03 (13.31) | | | |
| 12. Were you prevented from having immediate contact with the child before he/she was taken away for check-ups? [¿se te impidió el contacto inmediato con el niño o niña antes de que se lo llevaran para los controles?] | 18.2 | 27.72 (20.44) | −4.62 | 51.27 | p<0.001 |
| | 81.8 | 12.56 (14.26) | | | |
| 13. Were you made to feel guilty about any negative birth outcomes? [¿Te hicieron sentir culpable por algún resultado negativo del parto?] | 6.8 | 39.76 (19.33) | −6.91 | 234 | p<0.001 |
| | 93.2 | 13.41 (14.79) | | | |

*Note.* The Birth Conditions questionnaire items were administered in Spanish as designed by Ramos and Avila [38]. The research team performed the English translation of these items for the publication of this paper. They are not empirically validated in English version.

**Perceived social support.** Multidimensional Scale of Perceived Social Support (MSPSS, [42]). The MSPSS scale assesses the person's perceived social support at the global level, also consists of three subscales: perceived social support from family, from friends, and from a significant third party. It comprises 12 items (4 for each subscale) and a Likert-type response scale (7 response options). The present scale has been validated in Spanish [43] and present some

evidence of cross-cultural validation [44] although it could be improved; however, it has not been specifically validated with a postpartum population. In the present study, the instrument presented a reliability index of α = .94 for the total scale, for the Friends subscale; α = .95 for the Family subscale; α = .93, for the Significant Other subscale; α = .92. The scores from these three subscales were used as continuous variables.

## Data analysis

Statistical analyses were performed using IBM SPSS Statistics version 22 software. The significance level of $p < 0.05$ was considered statistically significant, while $p < 0.01$ was used to indicate stronger evidence of association. Initially, descriptive analyses of socio-demographic data and psychological measures of the participants were performed. The normality of the dependent variable (P-PTSS) was assessed using the Shapiro-Wilk test ($W = 0.98$, $p = 0.07$), indicating no significant deviation from normality. Additionally, visual inspection of histograms and Q-Q plots showed an approximately normal distribution. The skewness (−0.45) and kurtosis (0.32) values fell within the acceptable range for normality (−1 to +1). These results confirm that P-PTSS follows an approximately normal distribution, supporting the use of parametric statistical analyses

Subsequently, Pearson bivariate correlation analyses were carried out on the association between risk factors and P-PTSS. Moreover, t-tests for independent samples were carried out for differences in P-PTSS by other risk dichotomous variables (such as indicators of OV). The criteria of Cohen [45] were used for the interpretation of the magnitudes of the correlations. Finally, to explore the association between risk and protective factors and the dependent variable (P-PTSS), a three-step linear regression analysis was conducted. The model examined the contribution of pregnancy-related factors (Step 1), childbirth-related factors (Step 2), and perceived social support (Step 3), allowing for an exploratory assessment of their potential influence on P-PTSS. To assess the association between each independent variable and the dependent variable (P-PTSS), we used the standardized beta coefficient (β) from the regression model. A positive β value indicates that higher levels of the independent variable are associated with higher P-PTSS scores, whereas a negative β value suggests an inverse relationship. Statistical significance was determined using p-values, with thresholds set at $p < 0.05$ and $p < 0.01$. To evaluate the explanatory power of the model, we used the adjusted $R^2$ coefficient, which represents the proportion of variance in P-PTSS explained by the independent variables while accounting for the number of predictors included in the model.

## Results

Our study provides valuable insights into the relationship between obstetric violence (OV) and perinatal post-traumatic stress symptoms (P-PTSS), identifying specific childbirth experiences that contribute to the development of P-PTSS. We identified three key aspects of the birth experience that serve as risk factors of P-PTSS: feeling disqualified or mocked by healthcare staff during labor, lack of information about medical procedures, and being made to feel guilty by health professionals about a negative childbirth outcome. Furthermore, we highlight two protective factors: skin-to-skin contact in the immediate postpartum period, and perceived social support from family and friends.

The descriptive data on the participants' experience of in the study are shown in Table 1. At four to six weeks postpartum, 17% of women met the criteria for a clinical level of P-PTSS, as they endorsed the minimum required number of items in each of the four symptom clusters (B to E), according to the scoring interpretation described in the Instruments section regarding the PCL-5 scale. Additionally, 10% of the sample exhibited a subclinical level of P-PTSS, meeting the item requirements in three out of the four clusters. However, at this point, the six-month duration criterion for a formal P-PTSS diagnosis had not yet been met. The majority of women, representing 74% of the sample, did not exhibit symptomatology consistent with P-PTSS.

Descriptive data for the dichotomous measures of the OV Questionnaire are shown in Table 2, and further mean and standard deviations of dimensional psychological measurements are shown in Table 3.

**Table 3. Mean and Standard Deviations of dimensional psychological measurements.**

| Variables | M (SD) |
|---|---|
| Posttraumatic stress symptom level | 15.00 (16.27) |
| Perceived social support from the family | 20.09 (5.02) |
| Perceived social support from friends | 19.39 (5.29) |
| Perceived social support from significant others | 21.34 (4.32) |

*Note. M=Mean; SD= Standard Deviation*

As shown in Table 2, the most prevalent OV indicator in our sample was the performance of some medical procedures without sufficient information, recorded in 63.6% of the subjects. Of these procedures, the most prevalent were intravenous line placement (15.3%) and episiotomy (12.3%).

### Bivariate correlations and T-tests for independent samples

Bivariate correlation analyses showed significant correlations between P-PTSS and perceived social support. P-PTSS was significantly and negatively moderate related to perceived social support from friends (r = −0.375; p = 0.001), from family (r = −0.350; p = 0.001), and low from a significant other (r = −0.244; p = 0.001).

The t-tests for independent samples (Table 2) indicated that women who endorsed any of the items from the childbirth conditions questionnaire showed higher perinatal post-traumatic stress symptomatology, in comparison with those who don´t.

### Linear regression model considering three-step factors related to P-PTSS

A linear regression analysis was conducted to explore the association between risk and protective factors and the dependent variable (P-PTSS), assessed between the fourth and sixth week postpartum. The model was built in three sequential steps to evaluate the incremental contribution of different sets of predictors: Step 1: Pregnancy-related factors – This step included medical complications during pregnancy as a predictor variable; Step 2: Childbirth-related factors – In this step, additional predictors were incorporated, including the type of childbirth (categorical variable), the presence of medical complications during delivery (dichotomous variable), indicators of obstetric violence (13 dichotomous variables), and the presence of skin-to-skin contact after birth (dichotomous variable); Step 3: Postpartum social support factors – The final step introduced perceived social support from friends, family, and a significant other (three continuous variables).

Each step allowed us to assess the incremental variance explained by each set of variables. The adjusted $R^2$ coefficient was used to evaluate the explanatory power of the model, and standardized beta coefficients (β) were reported to determine the strength and direction of the association between each predictor and P-PTSS. The final model results are presented in Table 4.

As shown in Table 4, the model accounted for 49.9% of the variance in P-PTSS between the fourth and sixth weeks postpartum. On the one hand, medical complications during pregnancy (β = 0.03, p = 0.62) did not show a statistically significant effect, indicating that these complications alone are not risk factors of developing P-PTSS. On the other hand, childbirth-related variables explained 41.5% of the variance in P-PTSS. Specifically, three indicators of obstetric violence were significant risk factors of developing higher P-PTSS scores: feeling disqualified or mocked by healthcare personnel during labour (β = 0.15, p = 0.02); lack of information about medical procedures (β = 0.12, p = 0.03); being made to feel guilty about negative childbirth outcomes by healthcare staff (β = 0.17, p = 0.004). These findings suggest that perceived mistreatment during childbirth is strongly associated with increased postpartum traumatic stress symptoms. Finally, postpartum factors explained an additional 8.4% of the variance in P-PTSS, with two variables emerging as significant protective factors: skin-to-skin contact was negatively associated with P-PTSS (β = −0.17, p = 0.008), indicating that immediate

**Table 4. Linear regression model of the dependent variable perinatal post-traumatic stress symptoms.**

| Variables | R | R² | Adjusted R² | SE | Sig. |
|---|---|---|---|---|---|
| Pregnancy (1) | 0.02 | 0.01 | −0.01 | 16.30 | 0.706 |
| Childbirth (2) | 0.67 | 0.45 | 0.42 | 12.45 | p<0.001 |
| Postpartum (3) | 0.73 | 0.54 | 0.50 | 11.52 | p<0.001 |
| Variables | | | | β | t |
| Problems during pregnancy | | | | 0.03 | 0.62 |
| Type of delivery | | | | 0.07 | 1.30 |
| Problems during childbirth | | | | 0.03 | 0.66 |
| Item 1: Did you feel that your labor was sped up or slowed down for the convenience of the healthcare staff? | | | | 0.03 | 0.55 |
| Item 2: Did trainees intervene during labor or childbirth without your consent? | | | | 0.01 | 0.21 |
| Item 3: Did you feel that you were being infantilized (through the use of diminutives, nicknames, by making decisions without informing you, oversimplifying explanations, etc.) as if you couldn't understand what was happening or express your opinion? | | | | 0.03 | 0.57 |
| Item 4: Did you find asking questions or expressing your fears or concerns difficult or impossible because you were not answered or answered badly? | | | | 0.01 | 0.14 |
| Item 5: Did you feel neglected by the healthcare staff during labor? | | | | 0.02 | 0.39 |
| Item 6: Were you forced to stay in bed, prevented from walking or changing position? | | | | −0.02 | −0.37 |
| Item 7: Do you think the healthcare staff made ironic, disparaging, or mocking comments about your behavior or feelings? | | | | 0.15 | 2.40* |
| Item 8: Were you prevented from expressing your emotions (crying, screaming, laughing, etc.) during labor or childbirth? | | | | 0.09 | 1.57 |
| Item 9: Please let us know if any of the following procedures were carried out without providing you with sufficient information | | | | 0.12 | 2.19* |
| Item 10: Did you feel threatened or insulted during labor or childbirth? | | | | 0.01 | 0.21 |
| Item 11: At the time of delivery, were you prevented or hindered in your choice of birthing position? | | | | 0.09 | 1.52 |
| Item 12: Were you prevented from having immediate contact with the child before he/she was taken away for check-ups? | | | | 0.06 | 1.13 |
| Item 13: Were you made to feel guilty about any negative birth outcomes? | | | | 0.17 | 2.97** |
| Skin-to-skin contact | | | | −0.17 | −2.67** |
| Support from friends | | | | −0.26 | −3.95** |
| Support from family | | | | −0.17 | −2.43* |
| Support from a significant other | | | | 0.12 | 1.70 |

*Note.*

(1): medical problems during pregnancy;

(2): type of childbirth, presence of medical problems during childbirth, indicators of obstetric violence, and skin-to-skin contact;

(3): perceived social support from friends, family, and significant other;

*Regression coefficient significant at p<0.05

**Regression coefficient significant at p<0.01

physical bonding with the newborn may help reduce posttraumatic stress symptoms; and perceived social support from friends and family also showed a negative association with P-PTSS (β = −0.26, p < 0.001 and β = −0.17, p = 0.02, respectively), suggesting that a supportive social environment plays a key role in mitigating postpartum psychological distress.

## Discussion

The aim of the present study was to examine the association between the perception (or absence) of indicators of obstetric violence (OV) during childbirth and the occurrence of postpartum post-traumatic stress symptoms (P-PTSS), in conjunction with other potential risk and protective factors such as childbirth experience, initiation of early skin-to-skin contact

with the baby, and perceived social support during the postpartum period. To achieve this, we gathered data from a sample of 236 mothers, during their 4–6-month postpartum period.

Our results indicate that 17% of women in our sample met the criteria for a clinical level of P-PTSS, and 10% exhibited a subclinical level of P-PTSS.These results are comparable to the prevalence of PTSD in the general population [46] during the COVID-19 pandemic [47], and higher than prevalences found in other studies [48,49]. This speaks to the fact that, despite the existence of most women who seem to undergo good experiences of childbirth, there is a significant proportion of mothers who are partially traumatized by their experience of giving birth. This may negatively affect their mental health, their transition to parenthood and, in the mid to long term, their children's well-being.

Our results indicate a potential relationship between P-PTSS and experiences of OV. Specifically, our findings suggests that certain events during the birth experience may act as risk factors for the development of P-PTSS. First, women who reported feeling disqualified or mocked by health care staff during labor appeared more likely to develop P-PTSS. Secondly, lack of information about the performance of certain medical procedures emerged as a possible risk factor.. Thirdly, some mothers in our sample described feeling guilty due to comments from health professionals about a negative childbirth outcome, which might be associated with P-PTSS development. These results should be taken with caution, due to limitations of this research (see below for details), particularly the fact that reports about the experience of childbirth (and obstetric violence) were gathered from memory, sometime after the mother had undergone labour.

These findings may have relevant implications. First, the quality and humanity of hospital services—particularly in obstetric care—could be related to mothers' well-being. Providing adequate information on childbirth procedures, responding to mothers' needs during childbirth and training health professionals to be aware of factors contributing to negative appraisals of childbirth are essential to reduce fear of childbirth and prevent P-PTSS symptoms in the puerperium [6]. Furthermore, other studies suggest that this influence extends to the transition to motherhood, as previous literature has indicated a link between P-PTSS, disrupting breastfeeding [25], maternal-infant bonding [25–27]. This makes the quality of obstetric care a health issue for both women and children, as previous research also shows that bonding difficulties can lead to developmental disturbances [29,30,50] and emotional, social and relational problems in children [31,32].

Second, a more precise look should be taken at the practices of health personnel that, without being physically or obviously violent, may contribute to experiences of emotional pain, involving feelings of helplessness, shame, or guilt for women. One of the factors most strongly related to PTSS in humans is shame [19,20]. Our findings suggest that shame may play an important role in P-PTSS [20]. Among the women in our sample, traumatization did not necessarily respond to threats affecting the physical integrity or survival of the mother or the baby, but may have stemmed from experiences of inferiority, inadequacy, a sense of loss of dignity or the experience of being treated as "less than human". This causation has been described for other types of victimization and PTSD [20] and is in line with previous research that points to the central relevance of women's subjective experiences above the factual events of the childbirth process [18].

It is therefore vital to continue working on the identification and assessment of women's mistreatment during childbirth by health personnel, in order to obtain more accurate prevalence rates of the suffering of OV, as well as better strategies to eliminate this type of violence [51].

In addition to identifying risk factors for the development of P-PTSS during early postpartum, our results indicate the importance of various protective factors against the development of this symptomatology. On one hand, concerning the childbirth experience, providing women with the opportunity for skin-to-skin contact with their babies in the hours following birth would be a protective factor, buffering the negative impact of potentially traumatic births and preventing the onset of P-PTSS in the short to medium term. Previous research has also found that skin-to-skin contact reduces the chances of developing post-traumatic stress, for example, after caesarean sections or births classified as traumatic. These protective effects likely stem from a series of psychophysiological mechanisms associated with skin-to-skin contact, such as reduced cortisol levels and increased oxytocin in women's brain circuits. The latter would result in decreased feelings of fear or guilt and increased well-being. Additionally, skin-to-skin contact is related to other parenting processes such as success in

breastfeeding or the development of a strong bond with the baby, which overall increase the mother's sense of confidence and well-being.

Our research also highlights the important role of the social environment for the mother during the postpartum period, with perceived social support from both family and friends reducing the likelihood of post-traumatic stress within weeks of giving birth.This result is in line with previous findings in the PTSD literature [12,52], and probably is related to the power that positive social connections have in counteracting feelings that are central to post-traumatic conditions, such as fear, isolation, and the shame [20]. In trauma literature, it is argued that trauma impairs our psychological well-being by damaging our social sense of identity and our relationships with our environment, others, or the world at large [53]. Recovery or resilience, accordingly, occurs when our sense of identity and connection with others is restored or strengthened. This concept is known as the "social causation" theory, where social resources (such as social support) predict well-being, and their absence leads to psychological distress [54]. This relationship persists even in the perinatal context [9].

## Limitations

This study is not without limitations. Firstly, the educational level of the participants is generally favourable, which limits the generalizability of our results. Previous studies suggest that individuals facing vulnerability or social exclusion are more susceptible to experiencing traumatic events and developing post-traumatic conditions [55,56], implying that they may also be impacted differently by OV. Future research should analyse large samples with greater diversity in socioeconomic conditions, among other factors.

Second, the evaluation of OV has been done through single indicators, generated by Ramos and Ávila [38] through inputs from perinatal professionals. Although these indicators reflect very similarly aspects defined by the WHO [17] as OV, they could be improved. Specifically, we believe that the formulation of some items could benefit from a higher degree of specificity regarding the violent dimension (i.e., not informed or medically indicated) of certain measures. For example, items such as "Requirement to stay in bed without being able to walk or change position" would require additional information to be considered an indicator of OV (e.g., mother was told to remain in bed without the measure appearing to be medically indicated). This item should be accompanied by information on the presence or absence of analgesia methods (and the freedom or lack thereof in their choice) to assess whether their corresponding consequences, such as difficulty or impossibility of movement in some situations, have been reported. Similarly, items such as "Prevention or difficulty in choosing the birthing position" or "Prevention of immediate skin-to-skin contact before performing checks" should a priori be accompanied by similar expressions such as "without a proper explanation of the medical reasons justifying it." Nevertheless, it is considered a first approximation to the construct, as there are currently no self-reported measures, in the Spanish-speaking context, regarding the experience of OV during labour, from the mother's perspective. Finally, the social support scale consists of a subscale called "significant other," with items asking if the person has a special individual supporting them. When designing the survey, participants were not asked to specify who they were thinking of when completing these items. Therefore, scores from the family or friend's subscales may overlap, as a family member or friend may be the same person respondents indicated in the "significant other" subscale items. We believe that this study highlights the relevance that some of these actions may have on the subsequent development of maternal pathology and, therefore, the need for further research on this topic. In this regard, future studies should strive to deepen the assessment of OV as a global construct through the Birth Experience Questionnaire [54].

## Future lines

Our results are aligned with a series of recent studies that highlight the importance of emotional care for mothers during childbirth and their first contact with their baby, as well as the need to protect the emotional well-being of healthcare professionals so they can provide adequate support to mothers during this vulnerable and significant moment. In future research within this emerging field, a focus on resilience factors among mothers and babies exposed to OV is essential.

In our sample, 17% of mothers exposed to OV indicators exhibited posttraumatic symptom levels, while another 10% displayed subclinical symptoms, comparable to the non-exposed group, suggesting that not all women exposed to perinatal threats are significantly affected by posttraumatic stress. Exploring the processes related to good trajectories of perinatal mental health in mothers facing emotionally challenging deliveries should be systematically studied, to achieve a complex understanding of these phenomena and inform educational or preventive interventions. Additionally, research should trace the causal pathways from OV to infant development quality through mothers' mental health and postnatal bonding difficulties. Finally, continuing to delve into an ideographic approach, employing qualitative analysis, can unveil the subjective experiences of childbirth for these mothers, further exploring the significance of medical practices, information provision, and the attitudes of healthcare personnel during delivery [55,56].

As a diversity of studies from the the scientific suggest, the risk factors for experiencing childbirth as traumatic are diverse and can stem from both medical interventions [57] and the social and professional environment [58]. Stress induced by potential interventions during childbirth, fear for one's own or the baby's life, early separation of the mother-baby dyad, and the presence of OV by healthcare personnel are some of the elements that may contribute to this experience [57]. Additionally, current research points towards risk factors that contribute to increased professional burnout in obstetrics, which is related to a decrease in the quality of care for women giving birth [58]. Studies indicate that most healthcare professionals show a high risk of compassion fatigue, moderate risk of burnout, and low levels of satisfaction with the compassion they provide [58]. Additionally, exposure to traumatic births affects the psychological well-being of midwives to the extent of experiencing Burnout Syndrome [58]. It would be reasonable to think that the potential adverse consequences of this syndrome may threaten the provision of quality maternal care, as healthcare professionals' could distance themselves from patients as a means of self-protection, resulting in low satisfaction with the care they provide, as suggested by empirical evidence [58]. Additionally, significant stress derived from increased supervision and bureaucracy associated with patient care, teaching, and research; decreased reimbursement for patient care services; and reduced time available for teaching and research also influence the quality of obstetric care [59], thus impacting mothers' experiences during childbirth. Therefore, it is essential to direct research efforts towards identifying factors contributing to both traumatic childbirth experiences and the emergence of less empathetic and careful practices in obstetrics.

## Conclusions

The present study contributes to highlighting the relationship between indicators of OV and P-PTSS. It underscores the relevance of practices associated with shame and helplessness more than with a threat to physical integrity. Simultaneously, the study points to the potentially protective power of early skin-to-skin contact and the perception of social support.

These results provide valuable insights and can pave the way for upcoming longitudinal studies to further explore the intricacies of perinatal mental health. Replicating and expanding on the observed associations will enhance our understanding of risk factors across the mothers' life trajectory, pregnancy, and their impact on childbirth experiences, subsequent postpartum mental well-being, the quality of the breastfeeding, mother-child bonding, and child development.

## Author contributions

**Conceptualization:** Maria Vega-Sanz, Amaia Halty, Carlos Pitillas.

**Data curation:** Maria Vega-Sanz.

**Formal analysis:** Maria Vega-Sanz.

**Funding acquisition:** Ana Berástegui.

**Investigation:** Maria Vega-Sanz.

**Methodology:** Maria Vega-Sanz.

**Supervision:** Amaia Halty, Ana Berástegui.

**Writing – original draft:** Maria Vega-Sanz, Sofía Goñi-Dengra.

**Writing – review & editing:** Maria Vega-Sanz, Amaia Halty, Sofía Goñi-Dengra, Carlos Pitillas, Ana Berástegui.

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
