## [Decision Letter · Decision Letter 0]

3 Jan 2025

PONE-D-24-24985Exploring Traumatic Childbirth: Associations between Obstetric Violence indicators and Posttraumatic StressPLOS ONE

Dear Dr. Vega Sanz,

Thank you for submitting your manuscript to PLOS ONE. After careful consideration, we feel that it has merit but does not fully meet PLOS ONE’s publication criteria as it currently stands. Therefore, we invite you to submit a revised version of the manuscript that addresses the points raised during the review process.

We look forward to receiving your revised manuscript.

Kind regards,

Flávia L. Osório, PhD

Academic Editor

PLOS ONE

Journal Requirements:

2. Thank you for stating the following financial disclosure: Universidad Pontificia Comillas 

3. In the online submission form, you indicated that the dataset analyzed during the current study is available from the corresponding author on reasonable request.

Maria Vega-Sanz

mvsanz@comillas.edu

Reviewers' comments:

Reviewer's Responses to Questions

**Comments to the Author**

1. Is the manuscript technically sound, and do the data support the conclusions?

Reviewer #1: Yes

Reviewer #2: Partly

2. Has the statistical analysis been performed appropriately and rigorously? 

Reviewer #1: Yes

Reviewer #2: I Don't Know

3. Have the authors made all data underlying the findings in their manuscript fully available?

Reviewer #1: Yes

Reviewer #2: Yes

4. Is the manuscript presented in an intelligible fashion and written in standard English?

Reviewer #1: Yes

Reviewer #2: No

5. Review Comments to the Author

Reviewer #1: First, I would like to thank the authors and the journal for the opportunity to review the manuscript.

-The topic seems to meet the journal criteria.

-The introduction is sufficient, however, I´ve got a few comments to share with the authors:

1st. The concept of Perinatal post-traumatic stress symptoms (P-PTSS) is not defined. Considering the wide amount of research about the topic that can be found on PubMed, some readers may misunderstand the concept and confound it with Postpartum post-traumatic stress symptoms or even disorder. This must be clarified. Defining also perinatal period can help in this matter.

2nd. The concept of OV is presented first in a simple way, but after, the authors develop a complex definition and I think the first paragraph of the sub-theme should be deleted and the create a new introductory paragraph considering all the concept dimensions.

Also, I think this part of the paper will benefit from some restructuring as is a little bit "messy" when reading it, some lines are a collection of factors and are hard to read without losing the train of thought.

In addition, at least one contrasted dimension not mentioned in this manuscript: psycho-affective needs to be addressed.

3rd. I´ve got problems finding the true aim or goal of the study. There are a few lines where the authors point out the aim pursued, but some of them are just either ambiguous or provoke some degree of contradiction. E.g.:

Lines 69-70 VS Lines 140-141. In fact in some parts of the manuscript seems like we are addressing P-PTSS as a result of OV, but in some others feels the other way round. Some readers may fall into the bidirectionality of the phenomena.

-Methods: Please add in table one also net numbers apart from %.

Furthermore, please explain whether the tools used and administered were validated in the population of the study or similar sample, this will help to draw stronger conclusions from the results. If so, add the cites.

In this part of the manuscript, my bigger concern flaws; that the authors used the Questionnaire on Birth Conditions. This questionnaire allows for the identification of some indicators of OV during childbirth (3,6,16,18); at the same time, it comprises 13 items. But who/how/when/where is supported that by selecting just these four items we could determine the presence of OV? If we consider the rule created by the authors "the items have been used as 13 independent indicators" including the procedures described and undertaken without information given; pretty much everything that the woman suffers, or even happens to her could be classified as OV (as an example an EMCS with GA ending with FFP or RBC transfusion, will possibly fall into this category; may the mum not remember the explanation given prior to the procedure and still a normal intervention in an emergency situation; thus this would not be appropriate). And I think this is the major problem of the study. I would recommend the authors to check this thoroughly. While all my other comments can be classified as minor adjustments, this is a major issue of the paper.

-Results: This seems clearly approached and well presented. I can't see any manipulation from tables and every test seems appropriate. However, I´m missing a paragraph where all the main findings are presented at the beggining of the section. Just to make clear what to add to the current knowledge.

-Discussion: Seems properly addressed. But I would recommend to rewrite more cautiously on page 6, lines 11-24. Given the limitations of the study, I would rather be more wary of making assumptions. The manuscript is not well numbered, so I refer to page 26.

-Future lines section: This section is correctly developed and well-written as well as Conclusions.

The references must be checked as there are some mistakes and duplicate citations. E.g.: 3,15,16,19, 23, etc.

Overall seems an interesting investigation and may be relevant. Honestly, after checking these comments I think the manuscript can be published. But I would like to see those adjustments undertaken.

Reviewer #2: The study provides an important analysis and debate on the relationship between obstetric violence and PTSD.

However, there are important points to be improved and clarified:

In all manuscripts, it is necessary to include most recent bibliographical references.

Method

- Include information on how the participants were recruited, where and how the questionnaire was administered.

- Include information and bibliographical references on the method section of the cross-cultural validation studies of the follow instruments: Questionnaire on Birth Conditions, PCL-5, Multidimensional Scale of Perceived Social Support for the Spanish population.

The note in table 2: ‘The Birth Conditions questionnaire items were administered in Spanish as designed by Ramos and Avila. The research team performed the English translation of these items for the publication of this paper’ should be inserted and better described in the method, in the part of the instrument to which it refers. The authors should describe how this process was carried out.

- All the statistical analyses employed require the dependent variable (PTSD) to be parametric (normal distribution). Using parametric statistical analyses on non-parametric data could compromise the validity of the results. Therefore, the authors need to include in the method how the normality of the dependent variable was tested and what the results were.

- The authors need to explain in more detail how the linear regression models were carried out; the way it is written is confusing and does not make it possible to understand the modelling process.

- The authors should include in the method the level of statistical significance adopted.

- The authors should include in the method which regression parameters were used and how they were interpreted: e.g., to assess the association between independent and dependent variables, which regression parameter was used?

To assess the explanatory power of the model, which parameter was used?

- All the variables used in the linear regression model must be better described in the method (how they were obtained, how they were measured).

- The authors should define whether they are building an explanatory (risk/protective factors) or predictive model (predictive factors), there is a difference between them and throughout the manuscript the authors use the two definitions (explanation and prediction) as synonyms.

- In table 4 all VO variables should be described in full and not just item 1, 2, 3...

- page 4 ‘showed significant positive predictive power.’ What do the authors mean by this? Based on which statistics? This interpretation is confusing.

- page 4 ‘Our results show that more than 15% of women in our sample suffered significantly from one or two types of childbirth-related posttraumatic symptoms, and 10% were close to clinical levels of posttraumatic childbirth-related symptomatology.’ These results should be described in the results section.

- Page 4 ‘clinical levels of posttraumatic childbirth-related symptomatology.’ It was not clear in the method how the authors made this classification.

- The authors use risk factors as a synonym for predictive factors. There is a difference between these two types of variables, the authors should define which one it is.

- The last paragraph of the conclusion is not based on the methods and results of the study, I suggest rewriting.

6. PLOS authors have the option to publish the peer review history of their article (what does this mean? ). If published, this will include your full peer review and any attached files.

**Do you want your identity to be public for this peer review?** For information about this choice, including consent withdrawal, please see our Privacy Policy .

Reviewer #1: No

Reviewer #2: No

---

## [Author Response · Author response to Decision Letter 1]

14 Mar 2025

1º Reviewer(s)' Comments to Author Manuscript in general

The concept of Perinatal post-traumatic stress symptoms (P-PTSS) is not defined. Considering the wide amount of research about the topic that can be found on PubMed, some readers may misunderstand the concept and confound it with Postpartum post-traumatic stress symptoms or even disorder. This must be clarified. Defining also perinatal period can help in this matter.

Thank you very much for your suggestion. We have clarified what is meant by the perinatal period and completed the definition of perinatal post-traumatic stress at the beginning of the manuscript. Please find attached the paragraph as it would appear after the changes:

“Childbirth can be experienced by women as a traumatic event that may result in perinatal post‐traumatic stress symptoms (P-PTSS). It is important to note that the perinatal period encompasses not only childbirth but also pregnancy and the immediate postpartum phase. Following the recognition that some experiences during pregnancy and childbirth are sufficiently traumatic to lead to clinical conditions of Post Traumatic Stress Disorder (i.e., PTSD) (1), the term “Perinatal Post Traumatic Stress Disorder” (P-PTSD) was introduced to describe these clinical conditions. P-PTSS refers to the development of posttraumatic symptoms derived from or related to events occurring throughout the perinatal period, and it is distinct from postpartum PTSD or symptoms that arise solely after childbirth. These symptoms often include feelings of fear, terror, hopelessness, or helplessness (1-5).”

The concept of OV is presented first in a simple way, but after, the authors develop a complex definition and I think the first paragraph of the sub-theme should be deleted and the create a new introductory paragraph considering all the concept dimensions. Thank you very much for your suggestion. We have rewritten the introduction to provide a more integrated definition of OV, addressing the key dimensions of this construct.

“Obstetric Violence (OV) encompasses a range of practices during childbirth that violate women's autonomy, dignity, and rights. It can manifest physically through medical interventions performed without consent or with insufficient information, and psychologically, through degrading or infantilizing treatment by healthcare providers (15). These experiences can lead to feelings of vulnerability, guilt, and loss of control, impacting maternal well-being and birth outcomes (3,5,16).

Physically, OV includes procedures carried out without proper consent or under explicit refusal, such as episiotomies, repeated vaginal examinations by different professionals, the Kristeller maneuver, medically unnecessary acceleration or inhibition of labor, and the forced adoption of specific birthing positions without clinical justification (16). The WHO does not recommend these practices due to their potential negative consequences, including chronic pain, dyspareunia, perineal trauma, and urinary or fecal incontinence (17).

Psychologically, OV occurs when women are subjected to dehumanizing, dismissive, or authoritarian behavior by healthcare personnel. This includes being ignored, treated with sarcasm, denied the right to express emotions, or made to feel guilty for childbirth (3,5,16). In this context, the psycho‐affective needs of women—such as emotional validation, empathetic communication, and a supportive environment—are severely compromised. This neglect intensifies feelings of vulnerability, loss of agency, and isolation, thereby exacerbating the adverse psychological effects and increasing the risk of developing perinatal post‐traumatic stress symptoms (P-PTSS). Additionally, a lack of respect for the woman's birth plan, as well as coercion in decision-making, further contributes to these lasting emotional consequences (18)”

Also, I think this part of the paper will benefit from some restructuring as is a little bit "messy" when reading it, some lines are a collection of factors and are hard to read without losing the train of thought. We sincerely appreciate the reviewer’s insightful comments, which have helped us improve the clarity and structure of our introduction.

We have added subtitles within the introduction to better organize the information and ensure a more logical flow of ideas. Additionally, we have restructured several paragraphs, dividing complex ideas into more readable sections to enhance clarity and readability.

In addition, at least one contrasted dimension not mentioned in this manuscript: psycho-affective needs to be addressed.

Thank you for your suggestion. We have expanded on the psychological dimension affected by the presence of OV by specifically highlighting its psycho-affective aspect. Please find attached the paragraph referring to this issue:

“Psychologically, OV occurs when women are subjected to dehumanizing, dismissive, or authoritarian behavior by healthcare personnel. This includes being ignored, treated with sarcasm, denied the right to express emotions, or made to feel guilty for childbirth (3,5,16). In this context, the psycho‐affective needs of women—such as emotional validation, empathetic communication, and a supportive environment—are severely compromised. This neglect intensifies feelings of vulnerability, loss of agency, and isolation, thereby exacerbating the adverse psychological effects and increasing the risk of developing perinatal post‐traumatic stress symptoms (P-PTSS). Additionally, a lack of respect for the woman's birth plan, as well as coercion in decision-making, further contributes to these lasting emotional consequences (18)”

3rd. I´ve got problems finding the true aim or goal of the study. There are a few lines where the authors point out the aim pursued, but some of them are just either ambiguous or provoke some degree of contradiction. E.g.:

Lines 69-70 VS Lines 140-141. In fact in some parts of the manuscript seems like we are addressing P-PTSS as a result of OV, but in some others feels the other way round. Some readers may fall into the bidirectionality of the phenomena. We appreciate your feedback and have carefully reviewed the introduction, clarifying the terminology and ensuring that the study’s objective is clear and consistent. We eliminate ambiguities regarding the directionality between OV and P-PTSS, ensuring coherence throughout the text.

Methods: Please add in table one also net numbers apart from %. Thank you very much for the invitation. We have included in Table 1, a new column (N) in which we have included the absolute value of each subject presenting each of the variables listed in the table. It would be as follows:

Table 1. General aspects of childbirth.

Total Sample (N=236)

Variable N %

Complications during pregnancy

Yes 62 26.3

Type of delivery

Natural 183 77.5

Scheduled cesarean section 7 3

Emergency caesarean section 46 19.5

Complications during delivery

Yes 53 22.5

Baby's gestational week at birth

Extremely preterm 1 0.4

Very preterm 1 0.4

Moderate preterm 13 5.5

Full term 221 93.6

Note. Extremely preterm = less than 28 weeks; Very preterm = 28 to 32 weeks; Moderate preterm = 32 to 37 weeks; Full term = equal to or more than 37 weeks.

Furthermore, please explain whether the tools used and administered were validated in the population of the study or similar sample, this will help to draw stronger conclusions from the results. If so, add the cites. Thank you for your valuable comment. The PCL-5, used to assess post-traumatic stress symptoms, has a validated Spanish adaptation (Blevins et al., 2015). Although it was originally developed for assessing trauma-related symptoms in the general population, it has been specifically validated for evaluating traumatic childbirth experiences (Arora et al., 2023). Furthermore, several studies have utilized this tool to assess the onset of post-traumatic stress symptoms related to childbirth (DeVita et al., 2023; Orovou et al., 2022; Vega-Sanz et al., 2024), instructing participants to complete the questionnaire with their childbirth experience in mind.

The social support questionnaire, Multidimensional Scale of Perceived Social Support (Zimet et al., 1988), has a Spanish adaptation (Landeta & Calvete, 2002), but it has not been specifically validated for use in the postpartum population. We have included it as such and provide the corresponding citation below.

Additionally, the questionnaire assessing childbirth conditions was specifically developed and validated for Spanish-speaking populations. It has been used for its intended purpose—to evaluate childbirth experiences, making it a suitable instrument for this study.

We have incorporated this clarification in the Instruments section of the Methodology, ensuring that the validation and appropriateness of the tools for this population are explicitly stated.

“P-PTSS. The Post-traumatic Stress Symptom Checklist (PCL-5, 40) comprises 20 items and a Likert-type response scale (5 response options). This scale has a validated Spanish adaptation and also evidence of its cross-cultural validation (41). Although it was originally developed for assessing trauma-related symptoms in the general population, it has been specifically validated for evaluating traumatic childbirth experiences (42). It was specified that mothers should complete this questionnaire based on their childbirth experience.”

“Perceived social support. Multidimensional Scale of Perceived Social Support (MSPSS, 43). The MSPSS scale assesses the person's perceived social support at the global level, also consists of three subscales: perceived social support from family, from friends, and from a significant third party. It comprises 12 items (4 for each subscale) and a Likert-type response scale (7 response options). The present scale has been validated in Spanish (44) and present some evidence of cross-cultural validation (45) although it could be improved; however, it has not been specifically validated with a postpartum population.”

In this part of the manuscript, my bigger concern flaws; that the authors used the Questionnaire on Birth Conditions. This questionnaire allows for the identification of some indicators of OV during childbirth (3,6,16,18); at the same time, it comprises 13 items. But who/how/when/where is supported that by selecting just these four items we could determine the presence of OV? If we consider the rule created by the authors "the items have been used as 13 independent indicators" including the procedures described and undertaken without information given; pretty much everything that the woman suffers, or even happens to her could be classified as OV (as an example an EMCS with GA ending with FFP or RBC transfusion, will possibly fall into this category; may the mum not remember the explanation given prior to the procedure and still a normal intervention in an emergency situation; thus this would not be appropriate). And I think this is the major problem of the study. I would recommend the authors to check this thoroughly. While all my other comments can be classified as minor adjustments, this is a major issue of the paper. Thank you very much for your comment. First of all, we would like to clarify that for the analysis, all 13 indicators of the questionnaire were used—not only four. The numbers (3, 6, 16, 18) refer to the bibliographic references that support the notion that the content of these 13 indicators can be considered OV.

We totally agree that there could be cases where an unforeseen, highly severe situation requires immediate medical intervention, in which providing enough information to the woman about the procedure would not be possible. To avoid overrepresenting this item, we decided to add a question to the original 13 items specifying the type of procedure performed without providing information. This is detailed in the instrument’s description. Nevertheless, your comment has allowed us to add a clarification in the description. It now reads as follows:

“Additionally, and to further clarify the information regarding the "procedures without information" item, exposure to the following procedures performed without providing information was assessed: vaginal exams; refusal to provide food; artificial rupture of membranes (AROM); episiotomy; instruction to always lie down during childbirth; intravenous catheter insertion; genital shaving; enema; abdominal compression (Kristeller maneuver).”

In this way, we can ensure that the procedures without information reported in our paper refer specifically to these procedures—and not, for example, to the administration of emergency general anesthesia.

Results: This seems clearly approached and well presented. I can't see any manipulation from tables and every test seems appropriate. However, I´m missing a paragraph where all the main findings are presented at the beggining of the section. Just to make clear what to add to the current knowledge. Thank you for your valuable feedback. We have incorporated an introductory paragraph at the beginning of the results section to emphasize the main contributions of the study. Below, we provide the revised version:

“Our study provides valuable insights into the relationship between obstetric violence (OV) and perinatal post-traumatic stress symptoms (P-PTSS), identifying specific childbirth experiences that contribute to the development of P-PTSS. We identified three key aspects of the birth experience that serve as predictors of P-PTSS: feeling disqualified or mocked by healthcare staff during labor, lack of information about medical procedures, and being made to feel guilty by health professionals about a negative childbirth outcome. Furthermore, we highlight two protective factors: skin-to-skin contact in the immediate postpartum period, and perceived social support from family and friends”.

Discussion: Seems properly addressed. But I would recommend to rewrite more cautiously on page 6, lines 11-24. Given the limitations of the study, I would rather be more wary of making assumptions. The manuscript is not well numbered, so I refer to page 26. We sincerely appreciate your valuable comments and recommendations. We have carefully revised the discussion section and have reworded the conclusions to adopt a more cautious tone, considering the study's limitations. Specifically, we have avoided making definitive causal claims and have emphasized the need for further research to confirm and expand our findings.

The references must be checked as there are some mistakes and duplicate citations. E.g.: 3,15,16,19, 23, etc. Thank you very much for your contribution. We have reviewed the references to ensure that there are no duplicates and have also corrected any formatting errors in the citations.

2º Reviewer(s)' Comments to Author

In all manuscripts, it is necessary to include most recent bibliographical references. Thank you very much for your suggestion. We have reviewed the manuscript and updated the references accordingly.

Include information on how the participants were recruited, where and how the questionnaire was administered. We greatly appreciate your contribution. We have completed the first paragraph of the Participants section, specifying how the dissemination was carried out, how the sample was obtained, and the way the participants completed the questionnaires. We attach below the revised paragraph:

“A non-discriminatory exponential chain demonstration (i.e., snowball sampling technique) was performed to recruit the sample (N=236). The dissemination of the questionnaires was carried out through social networks, and participants were recruited via these platforms. They completed the questionnaires online through a Microsoft Forms survey. Sample recruitment began on 7 February 2023 and ended on 5 May 2023. The inclusion criteria for the study were being a woman over 25 years of age, residing in Spain, and being between the fourth and sixth week postpartum.”

Include information and bibliographical references on the method section of the cross-cultural validation studies of the follow instruments: Questionnaire on Birth Conditions, PCL-5, Multidimensional Scale of Perceived Social Support for the Spanish population. Thank you very

---

## [Decision Letter · Decision Letter 1]

25 Apr 2025

Exploring Traumatic Childbirth: Associations between Obstetric Violence indicators and Perinatal Posttraumatic Stress

PONE-D-24-24985R1

Dear Dr. Sanz,

We’re pleased to inform you that your manuscript has been judged scientifically suitable for publication and will be formally accepted for publication once it meets all outstanding technical requirements.

Kind regards,

Flávia L. Osório, PhD

Academic Editor

PLOS ONE

Additional Editor Comments (optional):

Reviewers' comments:

Reviewer's Responses to Questions

**Comments to the Author**

1. If the authors have adequately addressed your comments raised in a previous round of review and you feel that this manuscript is now acceptable for publication, you may indicate that here to bypass the “Comments to the Author” section, enter your conflict of interest statement in the “Confidential to Editor” section, and submit your "Accept" recommendation.

Reviewer #1: All comments have been addressed

Reviewer #2: All comments have been addressed

2. Is the manuscript technically sound, and do the data support the conclusions?

Reviewer #1: Yes

Reviewer #2: Yes

3. Has the statistical analysis been performed appropriately and rigorously? 

Reviewer #1: Yes

Reviewer #2: Yes

4. Have the authors made all data underlying the findings in their manuscript fully available?

Reviewer #1: Yes

Reviewer #2: Yes

5. Is the manuscript presented in an intelligible fashion and written in standard English?

Reviewer #1: Yes

Reviewer #2: Yes

6. Review Comments to the Author

Reviewer #1: Having the opportunity to review the manuscript after major revision: I have to recognize the improvements made by the authors. Based on the recommendations, the authors have thoroughly considered the suggestions and have taken steps to address them, thereby enhancing the credibility and completeness of their manuscript. No further comments or issued arose.

Reviewer #2: No additional comments, I have already send my comments.

The manuscript is adequate to publish on plosone

7. PLOS authors have the option to publish the peer review history of their article (what does this mean? ). If published, this will include your full peer review and any attached files.

**Do you want your identity to be public for this peer review?** For information about this choice, including consent withdrawal, please see our Privacy Policy .

Reviewer #1: No

Reviewer #2: No

---

## [Editor Report · Acceptance letter]

PONE-D-24-24985R1

PLOS ONE

Dear Dr. Vega-Sanz,

I'm pleased to inform you that your manuscript has been deemed suitable for publication in PLOS ONE. Congratulations! Your manuscript is now being handed over to our production team.

Kind regards,

on behalf of

Dr. Flávia L. Osório

Academic Editor

PLOS ONE